# Species-Dependent Structural Variations in Single-Domain Antibodies

**DOI:** 10.3390/antib14040100

**Published:** 2025-11-25

**Authors:** Marta Baselga, Javier Sánchez-Prieto, Víctor Manuel Medina Pérez, Alberto J. Schuhmacher

**Affiliations:** 1Institute for Health Research Aragon (IIS Aragón), 50009 Zaragoza, Spain; javiersanchez@unizar.es (J.S.-P.); vmmedina@iisaragon.es (V.M.M.P.); 2Fundación Agencia Aragonesa para la Investigación y el Desarrollo (ARAID), 50018 Zaragoza, Spain

**Keywords:** single-domain antibody fragments, species-dependent, database, VHHs, nanobody, llamas, alpacas, dromedary camels, Bactrian camels

## Abstract

Background/Objectives: Single-domain antibodies (sdAbs) are derived from camelid heavy-chain antibodies (HCAb). Their small size, high stability, and ease of production, among other properties, makes them highly valuable in biomedical research and therapeutic development. Several sdAb-based molecules are currently progressing through clinical trials, highlighting their translational relevance. As sdAbs originate from HCAb of *Camelidae* family, they can originate from multiple species including *Vicugna pacos*, *Lama glama*, *Camelus dromedarius* and *Camelus bactrianus*. Although several reports and databases analyze the structure of sdAbs, comprehensive evaluations on species-dependent structural differences remain scarce. Methods: We assembled MO-IISA, an open-access curated database of sdAbs with known antigen targets by integrating six public resources (iCAN, INDI, SAbDab-nano, sdAb-DB, PLabDab-nano, NbThermo) under harmonized eligibility criteria. Results: The final dataset comprises 2053 sdAbs derived from llamas (*Lama glama*, n = 1316); alpacas (*Vicugna pacos*, n = 325), dromedary camels (*Camelus dromedarius*, n = 377) and Bactrian camels (*Camelus bactrianus*, n = 35). We quantified region lengths, amino acid frequency, and conservation/entropy across frameworks (FR1–FR4). The average length of all sdAbs was about 124 ± 8 amino acids, with minor interspecies differences. We observed a consistent enrichment of lysines in FR3 (and secondarily FR2) and cysteines primarily in FR1 and FR3, with non-canonical cysteines more frequent in Bactrian and dromedary sdAbs CDRs. CDR2 and, particularly CDR3, contributed most to inter- and intra-species variability, whereas FRs were highly conserved. Conclusions: Species-neutral framework constraints and species-tuned loop adaptations have practical implications for sdAb engineering, species selection, and conjugation strategies. These features are captured in MO-IISA, an open-access database of known-target sdAbs from different species.

## 1. Introduction

Immunoglobulin G (IgG) from camelid mammals (including dromedary and Bactrian camels, llamas, alpacas, vicuñas, and guanacos) can be classified into six isotypes, of which two are conventional Abs (IgG1a and IgG1b), and the remaining four are heavy chain antibodies (HCAbs;IgG2a, IgG2b, IgG2c, and IgG3) [1,2]. HCAbs were first discovered in camel serum in 1993 [3,4]. These antibodies lack light chains yet remain functional [4]. HCAbs constitute approximately 60–80% of total IgGs in the serum of dromedary and Bactrian camels, and the 25–50% llamas and alpacas [5,6,7]. HCAbs contain a single variable domain (VHH) capable of antigen recognition with an affinity comparable to that of conventional antibodies and single-chain variable fragments (scFvs) Figure 1a) [3,8]. These antibody fragments, also known as single-domain antibodies (sdAbs) or nanobodies^®^ (Nbs, Ablynx), represent some of the smallest antigen-binding molecules (12–15 kDa) retaining functional capacity [9]. Being approximately ten times smaller than traditional antibodies, sdAbs exhibit faster and deeper tissue penetration and clearance (renal vs. hepatic from conventional mAbs), and the ability to access epitopes that are sterically inaccessible to conventional antibodies. In addition, they display high stability across a wide range of different pHs and temperature conditions, are easily produced, and possess good solubility [10,11,12].

The discovery that the variable domain of HCAbs (VHH) could be isolated and expressed independently led to the development of sdAbs [13]. These sdAbs can be recombinantly produced in microbial expression systems. Following antigen (or antigen-expressing cells) immunization, B lymphocytes can be isolated from peripheral blood, and total mRNA is extracted and reverse-transcribed into complementary DNA (cDNA) to amplify the VHH regions [2]. These amplified regions are then cloned into expression vectors for surface display in systems such as phage, yeast or bacteria [14,15,16]. Ideally, the constructed library should exhibit a complexity of >10^7^ individual clones, with at least 70% containing complete VHH sequences [14].

Structurally, sdAbs domains consist of three hypervariable complementarity-determining regions (CDRs) flanked by four conserved framework regions (FR). The FRs fold into nine β-strands arranged into two β-sheets (one containing five-stranded and the other four), which are stabilized together by a disulfide bond between two conserved cysteine residues (Figure 1b). The CDRs of a VHH are arranged in three loops connecting the β-strands and are spatially clustered near the N-terminus [8,17]. These three hypervariable loops of the VHH domain (designated H1, H2, and H3), correspond to the complementarity-determining regions (CDR1, CDR2, and CDR3) defined by IMGT (ImMunoGeneTics) numbering system, a standardized framework for the precise structural and functional annotation of immunoglobulin and T-cell receptor variable domains [18]. Together, these loops form the antigen-binding surface. H1 and H2 (CDR1–2) exhibit relatively conserved lengths and orientations [8]. Unique conformational epitopes, such as cavities or unstructured protein regions, can be recognized with high affinity and stability primarily due to the especially the longer H3 loops (600–800 Å) [19,20]. Although the overall CDR architecture is conserved, CDR3 exhibits the highest variability among the three loops. CDR1 and CDR2 remain relatively constrained in length and sequence to preserve the VHH fold; however, CDR1 is frequently elongated compared with conventional VH domains. This moderate extension, rather than broad variability, contributes to expanding the paratope surface and compensates for the absence of light-chain-derived CDRs [8,17,21].

Camelid VHHs share high sequence homology to human VH3 [22], and can readily humanized to reduce immunogenicity for their use in biomedical applications. Nonetheless, they exhibit key structural differences between them. In the framework regions, for example, the FR2 segment of a conventional VH domain contains four conserved residues (V42, G44, L45, and W47, according to IMGT numbering), mainly hydrophobic, that interact with the VL domain to maintain the VH–VL pairing [23]. Because these VHH domains function as monomers, these hydrophobic residues are typically substituted with smaller or more hydrophilic amino acids (commonly F37, E44, R45, and G47, according to IMGT numbering), enhancing their solubility and structural stability in aqueous environments [11,24].

Another significant difference from conventional antibodies lies in the composition of the antigen-binding region, which in conventional antibodies comprises six CDRs (three from the VH and three from the VL domains) whereas in sdAbs, the paratope is formed solely by the three CDRs of the VHH. To compensate for the absence of VL-derived CDRs, the CDR1, and particularly the CDR3, of VHHs are elongated, thereby providing an expanded interaction surface [25] and adopting a more protruding conformation compared with the relatively flat paratope of conventional VH domains [25,26]. This unique topology allows VHHs to access recessed or concave epitopes that are typically inaccessible to conventional antibodies [8,17,24]. Additionally, VHHs frequently contains an additional intramolecular disulfide bond linking cysteine residues in CDR1 and CDR3, which stabilizes the extended and flexible CDR3 loop [8,27,28].

Notable interspecies variations have been reported in the structural organization of VHH domains among camelids. While all VHHs share a conserved β-sandwich framework stabilized by a canonical disulfide bond connecting two conserved cysteines located in FR1 and FR3, the presence and positioning of additional disulfide bonds vary across species. In llamas, the canonical disulfide bridge is less frequently observed, and the CDR3 region tends to be shorter compared to other camelids. Conversely, in dromedaries, it is described that approximately 10% of VHHs contain an additional disulfide bond linking a second cysteine within CDR3 to another cysteine located in the FR2 region (position 50) [8,28]. Furthermore, a subset of llama VHHs (and occasionally dromedary VHHs) form an extra disulfide linkage between CDR2 and CDR3 [8,27]. Despite this variability, it is reported that the cysteines responsible for forming these disulfide bonds in CDR1, CDR2, and FR2 are generally conserved among species, while the cysteine position within CDR3 remains highly variable [8,28,29]. These species-dependent disulfide configurations contribute to differences in CDR3 length, structural stability, and conformational flexibility, ultimately influencing the binding diversity and biophysical characteristics of VHHs among camelids (Figure 1b).

Given the growing biomedical relevance of these immune nanostructures, this study presents a comparative analysis of sdAbs structural features among the four species most used for the generation of immune libraries: llamas, alpacas, dromedary camels, and Bactrian camels. We corroborate previously described and uncover novel interspecies differences in their molecular organization and biology, providing new insights into factors that might influence their suitability for distinct biomedical applications.

Although several large-scale studies have analyzed VHH or sdAb sequences, most of them lack information regarding the specific antigen targets, making it difficult to establish relationships between sequence features and functional properties [30,31]. To systematically investigate these structural variations, we compiled sequence and structural data from publicly available sdAb repositories and constructed a harmonized dataset of known-target sdAbs for comparative analysis across camelid species, focusing on sequence and structural features that may affect their stability, solubility, and overall suitability for biomedical use. This comparative approach aims to identify species-specific VHH sequence patterns that could serve as molecular markers and inform the rational selection of camelid sources for specific biotechnological or therapeutic uses.

## 2. Materials and Methods

### 2.1. Data Collection and Eligibility Criteria

Data related to sdAbs were collected from several publicly available repositories that provide sequence-level, structural, or biophysical information. The following resources were included in this study:PLabDab-nano (Oxford Protein Informatics Group). A curated database integrating sdAbs sequences and structural information, designed to facilitate comparative analysis and antibody engineering [32].Structural Antibody Database (SAbDab-nano). A subset of the SAbDab database focused on sdAbs, which compiles high-quality structural and functional annotations from the Protein Data Bank (PDB) [33].Integrative Camelid Antibody (iCAN) database, which provides a comprehensive collection of camelid sdAbs along with the associated experimental metadata [34].Integrated Nanobody Database Initiative (INDI). A broad resource incorporating multiple sources of sdAb data. In this study, three subsets were incorporated: (i) AbGenbank-derived sequences, (ii) structure-based entries, and (iii) manually curated records [35].sdAb-DB. A dedicated repository of synthetic single-domain antibodies, containing sequence and antigen-specific annotations.NbThermo. A specialized dataset that reports sdAb thermostability values alongside sequence information [36].

Each dataset was downloaded in its original format (CSV, FASTA, PBD, or tabular text). Other known datasets such as NanoLas2 [37] were excluded due to the impossibility of bulk download. A set of mandatory fields was established to ensure compatibility and integration across datasets, including sdAbs identifier, species of origin and complete sequence or CDRs (1–3) and FRs (1–4). Metadata were extracted whenever available. Additional fields (such as DOI, PDB code, antigen type and antigen affinity) were considered desirable but optional and were incorporated when available.

### 2.2. Data Processing

All datasets were imported into a unified workspace using Python (version 3.11) and the Pandas library. A standardized processing pipeline was designed and implemented to harmonize heterogeneous sources, enforce inclusion criteria, and ensure consistency across the final dataset. The workflow comprised the following steps: initial merging and quality control, species harmonization, sequence verification and reconstruction, removal of duplicates, detection and removal of artificially ordered sequences, final dataset creation (Figure 2a).

*Initial merging and quality control*. All records were concatenated into a single data frame. Entries missing any mandatory fields (sdAbs identifier, species, complete sequences or FRs/CDRs) were removed at this stage. Records lacking only optional metadata, namely antigen description, affinity or DOI, were retained.*Species harmonization*. Species annotations varied substantially across datasets. Therefore, the nomenclature was standardized and consolidated into four categories: *Llama glama* (Llama), *Vicugna pacos* (Alpaca), *Camelus dromedarius* (dromedary camel), and *Camelus bactrianus* (Bactrian camel). Entries with ambiguous or generic (e.g., ‘camelid’ or ‘mixed library’) were excluded to prevent misclassification.*Sequence verification and reconstruction*. Where full-length sequences were unavailable, but all CDRs and FRs were present, complete sequences were reconstructed by concatenating these regions. Entries missing one or more CDRs were removed since they were considered indispensable for downstream analyses. Additionally, to ensure consistent structural representation across entries, a custom Python function was developed which integrates the AbNumber library, which employs ANARCI. This library employs OPIG/IMGT-style antibody numbering to derive FR and CDR boundaries from full-length sdAbs sequences. It can also reconstruct missing elements when sufficient annotations are available. Entries that could not be reconstructed were deleted (Figure 2b).*Removal of duplicates*. Exact duplicate entries were removed using the full-length sequence field as the primary key.*Detection and removal of artificially ordered sequences*. During the quality control process, an additional filtering step was implemented to identify sequences that were artificially ordered or non-biological. In some cases, we found that deposited sdAb entries contained amino acid strings in which the residues had been deliberately ordered (e.g., in alphabetical order) or presented as homopolymers (e.g., “AAAA…”). It is thought that these anomalies arise when authors intentionally obscure the true sequence for reasons such as intellectual property protection or confidentiality. To address this issue, a dedicated function was used to identify suspicious patterns, including:Alphabetically ordered sequences (e.g., “ACDEFGHIKLMNPQRSTVWY”), which are not compatible with genuine protein structures.Low-complexity or homopolymer strings, such as extended runs of a single amino acid (e.g., poly-A or poly-G).

Entries matching these criteria were removed from the final dataset to ensure that only biologically plausible sdAb sequences were retained for subsequent analyses.

### 2.3. Determination of FRs Conservation and Entropy

Sequence conservation was evaluated to quantify variability across FRs and CDRs, both globally and stratified by species. A set of custom Python functions were designed to clean sequences, pad and align them by region, compute consensus residues and entropy values, and tabulate results.

*Region-specific alignments*. Rather than applying a global multiple sequence alignment to entire sdAb sequences, each region (FR or CDR) was analyzed independently. Sequences were pre-processed to retain only canonical amino acids, with any ambiguous characters replaced by a gap symbol (“–”). Within each region, the sequences were padded to the length of the longest entry to form a rectangular alignment matrix. This approach avoided introducing artifacts from conventional MSAs in highly variable loops, especially in the CDR3 region.

Per-column statistics. For each alignment column j

Let $n_{a,j}$ be the count of aminoacid a (excluding gaps) and $N_{j}=\sum_{a}n_{a,j}$.

The consensus residue at position j was defined as the amino acid with maximum frequency (Equation (1)):(1)$$Consensus\left(j\right)=argmax(n_{a,j}),$$

The consensus frequency was the relative proportion of that residue (Equation (2)):(2)$$ConsFreq\left(j\right)=\;argarg\underset{a}{max}\;n_{a,j}\;$$

To quantify positional diversity, the Shannon entropy was computed to each position (Equation (3)):(3)$$H_{j}=-\sum_{a}\frac{n_{a,j}}{N_{j}}\left(\frac{n_{a,j}}{N_{j}}\right),$$

Entropy values were expressed in bits, ranging from 0 (perfect conservation) to ${log}_{2}\approx 4.32$ (maximal diversity over 20 residues).

*Species-specific analyses*. The same analytical pipeline was applied to species-specific subsets of the dataset corresponding to the four species. This enabled identification of both conserved motifs shared across species as well as species-specific sequence variations.

### 2.4. Analysis of the Amino Acids

The amino acid composition was analyzed to characterize residue usage within sdAb sequences and to identify region- and species-specific biases. Particular emphasis was placed on residues that are important for structure or function, such as cysteine (C) and lysine (K), as they are frequently used for bioconjugation approaches.

For each sequence, the number of amino acids was tallied across the full sdAb sequence. These counts were then aggregated over the deduplicated dataset and normalized by the total number of residues to obtain the relative frequencies (Equation (4)):(4)$$f_{a}=\frac{\sum_{i=1}^{N}n_{a,i}}{\sum_{i=1}^{N}L_{i}}$$
where $n_{a,i}$ is the count of amino acid a in sequence i, $L_{i}$ is the sequence length, and N is the number of sequences. Frequencies were reported as percentages (100 $f_{a}$).

*Region-specific composition*. Using the IMGT segmentation obtained in Section 2.2, residue frequencies were computed separately for each framework (FR1–FR4) and complementarity-determining region (CDR1–CDR3). This allowed the identification of compositional trends associated with structural stability (frameworks) and antigen recognition (CDRs).

*Species-specific comparisons*. Equivalent frequency profiles were generated for subsets corresponding to each camelid species (*Lama glama*, *Vicugna pacos*, *Camelus dromedarius* and *Camelus bactrianus*), enabling the comparison of conserved versus divergent amino acid usage patterns across taxa.

The analysis focused on cysteine and lysine residues, as these amino acids are commonly used in conjugation strategies:Cysteine (C): counts and frequencies were recorded both globally and by region to detect canonical cysteine positions and potential non-canonical cysteine insertions that might form additional disulfide bonds.Lysine (K): counts were computed per region to evaluate the distribution of positively charged residues, which may influence antigen binding or overall molecular stability.

Data handling and visualization. Analyses were performed in Python (version 3.11) using the Pandas library for tabulation. Custom scripts iterated over sequences to quantify residue occurrences globally, by region, and by species. The resulting outputs included (i) frequency tables, (ii) bar plots of amino acid distributions and (iii) per-region cysteine/lysine histograms.

### 2.5. Statistical Analysis

Statistical analyses were performed using GraphPad Prism Software (v10.0, Boston, MA, USA). Data from Bactrian camels were not included in the statistical studies due to the small number of sdAbs from this species (n = 35). The Shapiro–Wilk test was applied to each group to assess the normality of the data distribution. For all groups, the null hypothesis of normality was rejected. Accordingly, non-parametric tests (Kruskal–Wallis or Mann–Whitney) were used to evaluate differences between groups. Statistical significance was defined when *p* < 0.05 (*), *p* < 0.01 (**), or *p* < 0.001 (***).

## 3. Results

### 3.1. MO-IISA: A Database of sdAbs with Known Targets

To systematically analyze interspecies structural variations, we collected sequential and structural data from publicly available sdAb repositories and developed a harmonized dataset for comparative analysis among camelid species. This effort focused particularly on differences that may influence their stability, solubility, and potential for biomedical applications. A total of 17,056 sdAbs were initially retrieved from 6 different databases (iCAN, INDI, SAbDab-nano, sdAb-DB, PLabDab-nano, and NbThermo). Following screening and application of the eligibility criteria, the dataset was refined to 2053 sdAbs (Figure 3a). These sdAbs sequences were retained because they contained a proper and complete sequence and had been validated or analyzed against an antigen. Accordingly, the resulting freely available MO-IISA database provides an updated collection of sdAb sequences with presumably known targets and other metadata of interest.

One of the objectives of this database is to enable the identification of potential sequence-structural characteristics associated with the species of origin. The final dataset includes sdAbs from four camelid species: alpaca (n = 325), Bactrian camel (n = 35), dromedary camel (n = 377), and llama (n = 1316) (Figure 3b). All databases contained sdAbs from alpaca, llama, and dromedary camel, except NbThermo, in which the sdAbs were originally labeled as ‘Camel’, preventing distinction between Bactrian or dromedary origins. In the case of Bactrian camels, all sdAbs were only collected from sdAb-DB (Figure 3c). Because the number of retrieved sdAbs recovered from Bactrian camels was small (n = 35), originated from a single database, and might represent samples from the same individual, data from this species were excluded from subsequent statistical studies.

Across species, all 20 amino acids are present in the sdAb sequences. Glycine (G), serine (S), and alanine (A) appeared most frequently. Conversely, cysteine (C), methionine (M), and histidine (H) were the least frequent (Figure 3d). These results quantitatively support previous observations that camelid VHHs exhibit subtle yet significant interspecies differences, particularly in CDR3 length and composition.

### 3.2. Framework Conservation and Entropy Are Not Species-Dependent

FRs were globally conserved across the MO-IISA database cohort (Figure 4a,b). The overall entropy and conservation data for all species can be found in Appendix A. No significant global differences between species were detected in either conservation or entropy (Kruskal–Wallis, *p* > 0.05).

As depicted in Figure 4c, FR4 was the most conserved region, displaying conservation values close to 1 in most positions and minimal entropy. FR1 and FR3 exhibited high conservation with low to moderate entropy, indicating that most positions were stable with few variable segments. In contrast, FR2 was the least conserved FR, characterized by moderate conservation and relatively higher entropy, reflecting greater sequence heterogeneity.

These results demonstrate that the framework architecture of camelid VHHs is highly conserved across species, with no significant differences in entropy or sequence conservation, supporting the structural stability of sdAbs irrespective of species origin.

### 3.3. CDRs from Alpaca-Derived sdAbs Are Smaller than in Other Species

The average length of sdAb sequences across all species was 123.8 ± 7.9 amino acids (aa), corresponding approximately to 13.6 ± 0.9 kDa, within a range of 10.45–18.7 kDa). The average length per species was 123.8 ± 9.0, 124.6 ± 5.1, 123.0 ± 5.4, and 124.0 ± 8.2 aa for alpacas, Bactrians, dromedaries, and llamas, respectively (Figure 5a). No significant differences were observed between species. The relative standard deviation (RSD) of the length was: 7.3%, 4.4%, and 6.7% in alpacas, dromedaries, and llamas; therefore, the length of the sdAb sequences of dromedaries is more homogeneous in length.

The lengths of the FRs did not vary either between species or within species, being ~25, 17, 38, and 11 aa in FR1, FR2, FR3, and FR4, respectively. However, CDRs varied between different species. The average length of CDRs was 8.5 ± 1.9 aa (CDR1), 8.0 ± 1.2 aa (CDR2) and 15.9 ± 4.3 aa (CDR3). The RSD of length was: 22.5%, 14.3%, and 27.1% in CDR1, 2, and 3. Therefore, CDR2 was the most homogeneous in length across all species.

The average CDR1s lengths were 8.4 ± 1.4, 7.8 ± 1.0, 7.8 ± 1.1 and 8.8 ± 2.1 aa in alpacas, Bactrians, dromedaries, and llamas, respectively (Figure 5b). CDR1s from llamas were significantly longer (*p* < 0.001) than that of alpacas and dromedaries. The RSD of the length was: 16.8%, 14.6%, and 24.3% in alpacas, dromedaries, and llamas. Therefore, llamas showed greater heterogeneity in CDR1 length.

Similarly, CDR2 lengths were significantly higher in llamas compared with alpacas and dromedaries (*p* < 0.001). However, the averages were similar (7.9 ± 1.1, 8.0 ± 1.0, 7.8 ± 0.7, 8.2 ± 1.3 aa in alpacas, Bactrians, dromedaries, and llamas) (Figure 5c). In addition, the RSD of length was: 13.3%, 9.4%, and 15.4% in alpacas, dromedaries, and llamas, indicating that dromedaries have more homogeneous CDR2.

CDR3 lengths exhibited the greatest variation both among and within species (Figure 5d). Alpaca-derived CDR3 were significantly shorter than those from dromedaries and llamas (*p* < 0.001). On average, CDR3 lengths were 15.2 ± 4.4 aa (alpaca), 17.2 ± 4.3 aa (Bactrian camel), 16.1 ± 4.3 aa (dromedary camel), and 16.0 ± 4.3 aa (llama). The RSD of the length was 29.1%, 26.7%, and 26.6% in alpacas, dromedaries, and llamas, indicating that the length of CDR3 are heterogeneous in all the species.

The amino acid composition profiles were comparable across all species, showing higher proportions of glycine (G), serine (S), and alanine (A), and a lower proportion of histidine (H), methionine (M), tryptophan (W), and cysteine (C) (Figure 5e).

These quantitative findings confirm that, although the overall VHH architecture is conserved among camelids, subtle but statistically significant variations occur across species, particularly in CDR3 length, refining and extending previous qualitative reports of interspecies structural diversity.

### 3.4. FRs Harbor Most Lysine and Cysteine Residues in Bactrian and Dromedary Camels

Previous structural studies suggested that the FRs of VHHs contain most of the conserved cysteine residues forming the canonical disulfide bridge, with occasional additional cysteines reported in certain camelid species. However, quantitative comparisons of cysteine and lysine residues distributions across species have remained limited. Analysis of the distribution of lysines and cysteines across the sdAb regions of camelid species shows highly conserved patterns. Lysine enrichment is observed in FR3 (and to a lesser extent in FR2), conserved across species, and cysteine distribution is limited to the structural positions of the native disulfide bridge.

Across all species, lysine residues were predominantly concentrated in FR3, with a secondary contribution in FR2, whereas the remaining regions (FR1, FR4, and CDR1–3) exhibited very low frequencies (Figure 6a). The heat map confirmed this pattern, with FR3 as consistently the most lysine-enriched region (Figure 6b).

Consistent with the canonical disulfide bond, cysteine residues were primarily located in FR1 and FR3, with values close to zero in the CDRs and in FR2/FR4 in Bactrians, dromedaries and llamas (Figure 6a). However, Bactrians and dromedary-derived sdAbs displayed a higher frequency of cysteines in CDR1 and CDR3 compared with other species, supporting previous reports of species-specific diversification in these regions.

## 4. Discussion

The VHH domains of HCAbs are highly homologous to the VH domains of conventional antibodies (showing high sequence homology to the human VH3 family) [22]. However, there are key differences between them. As previously reported, these differences mainly affect the FR2 region, residues influencing solubility, and the paratope configuration. Building upon this foundation, our work further explores how these molecular features vary across species using experimentally validated sdAbs.

Several sdAbs databases are available (including iCAN, INDI, SAbDab-nano, sdAb-DB, PLabDab-nano, NbThermo and NanoLas2) [32,33,34,35,36,37,38,39]. However, in most of these resources the sdAbs lack defined species of origin, the targeted antigen, and their sequences come from large-scale sequencing of immune libraries. Consequently, their immunoreactivity is unknown. To overcome these limitations, we compiled a novel curated database of sdAbs with known antigen targets, enabling more rigorous structural analyses grounded in sdAbs that are potentially functional. Initially, 17,056 sdAbs were collected, of which 2053 were retained after selection and application of the eligibility criteria. The database is freely available on the public GitHub repository (https://github.com/Javiersp4/MO-IISA accessed on 6 October 2025).

A total of 66.8% of the sdAbs included in the database originated from llamas, while the remaining 18.3%, 14.4%, and 0.5% were derived from alpacas, dromedary camels, and Bactrian camels, respectively. Given their greater accessibility, ease of handling, and extensive livestock and veterinary infrastructure across several countries, the preponderance of sequences and immune libraries from lamas and alpacas reflects an operational reality [15,40,41,42,43,44,45,46,47,48,49]. Compared with larger camelids (Bactrian and dromedary camels), the smaller size of llamas and alpacas facilitates housing, transportation, and biosecurity procedures. These species also tolerate repeated venipunctures and longitudinal sampling with minimal stress and are well suited for repeated immunization protocols [45]. These elements work in concert to produce a positive feedback loop that includes increased availability, more projects and data, improved protocols, and increased adoption. These logistical advantages, already recognized in the field, contributed to the dominance of llamas and alpacas in sdAb discovery pipelines [50,51,52,53,54].

The current consensus establishes a length for sdAbs of approximately 110–130 amino acids [55,56,57]. In agreement with this range, our analysis found that full-length sdAbs sequences average 124 amino acids (range: 95–170 aa). Median lengths by region were consistent with canonical sdAbs architecture. FRs were tightly distributed (FR1 ≈ 25 aa; FR2 ≈ 17 aa; FR3 ≈ 38 aa; FR4 ≈ 11 aa) and CDR1/2 were centered around 8 amino acids. All the CDRs of sdAbs derived from alpacas were significantly smaller than those of dromedaries and llamas. This difference was more pronounced in CDR3, where Bactrian camel VHHs (17 aa) display significantly longer but more narrowly distributed CDR3s than those of dromedary camels (16 aa), alpacas (15 aa) and llamas (16 aa). Analyses of CDR3s obtained through high throughput sequencing have reported similar average lengths across species [30,31]. Our findings are consistent with these prior reports and show that CDR3 exhibits the largest dispersion, characterized by a wide length distribution and small but consistent interspecies differences, with a tendency for shorter CDR3 in alpacas compared with llamas and dromedary camels. According to the literature, longer CDR3s are associated with lower net charge sdAbs, reduced surface charge, lower surface hydrophobicity, larger surface area, increased interactions between CDR3 and other regions of the VHH, and a smaller contribution of CDR1/CDR2/FR2 to antigen binding [28]. Our data support this view, highlighting how subtle CDR3 length variations may influence paratope configuration and antigen engagement.

The framework regions (FR1–FR4) do not diversify to generate specificity and remain conserved [58]. Consistent with this canonical model, our analysis revealed that the overall conservation of FRs is high, irrespective of species of origin. Most FR positions show conservation values between 0.8 and 1.0 (Figure 5), suggesting strong structural constraints on the fold. Our comparative analysis of camelid sdAbs revealed a highly conserved scaffold (particularly FR4) and variability mainly localized in FR2 and FR3. Overall patterns of amino acid composition and conservation or entropy metrics were strikingly consistent between species. These findings support the established notion that framework regions ensure proper folding and solubility, while CDR3 governs affinity and specificity by balancing structural stability with conformational flexibility. Beyond their structural roles, framework characteristics may also influence the suitability of sdAbs from different species for downstream applications. Subtle variations in framework residues, such as hydrophilic substitutions in FR2 and lysine enrichment in FR3, could affect expression efficiency and conjugation accessibility in biomedical applications.

FR2 represents a key region where sequence consensus varies between species and plays an important role in maintaining solubility and structural integrity. FR2 in VHH/sdAbs are enriched in hydrophilic residues, which, in contrast to conventional VHs, reduces nonspecific interactions with hydrophobic surfaces membranes and plasma proteins, thereby lowering off-target accumulation [59]. Our data aligns with this established mechanism, showing that FR2 remains conserved, exhibits a limited length, a low frequency of hydrophobic residues, and a moderate presence of lysine residues (positive charge and high solubility), features that are likely reduce systemic stickiness. This in turn results in more consistent pharmacokinetics and an improved signal-to-background ratio.

The major contribution to paratope recognition is consistent with the strong conservation of FRs (including the structural role of FR2 [60]) and the diversity of CDR3 [19,20]. By demonstrating that CDR2 maintains a limited length and that FR4 is the most conserved framework across species, our data expands upon this paradigm and suggest the presence of additional structural constraints within that loop. Taken together, the results corroborate and refine the hypothesis that the camelid VHH repertoire converges toward controlled variation and functionally equivalent architectures.

An exposed and charged surface that may promote solubility, expression, and site-selective conjugation is suggested by the enrichment of lysine residues in FR3. Although the additional disulfides reported in VHH subsets that stabilize long CDR3s are consistent with the moderate presence of cysteines in CDR1 and CDR3, their frequency and distribution should be examined individually to prevent undesired dimerization. This aligns with earlier reports describing variability in auxiliary disulfide bonds among camelids. Because they are species-neutral features, these chemical signatures provide predictable points of intervention for manufacturing and protein-format engineering. Solvent-exposed lysines in sdAbs offer convenient handles for amine-directed chemistries (e.g., NHS esters), whereas cysteines enable thiol-selective reactions (e.g., maleimides or disulfide re-bridging), together allowing robust, and often site-selective, conjugation for labeling, PEGylation, and payload attachment without disrupting the sdAb fold [61]. The predominance of lysine residues in FR3 implies that reactions with NHS esters will occur mainly in this region and, to a lesser extent, in FR2. This facilitates high yields but introduces species heterogeneity (Drug-to-Antibody Ratio [DAR] and positional microheterogeneity), with potential functional impact if any modified lysine located near the base of CDR3. These compositional biases are not merely sequence curiosities but define potential chemical handles for sdAb functionalization. Lysine-rich FR3 regions provide accessible amine groups for covalent labeling, whereas cysteine-containing CDRs or FRs enable thiol-specific conjugation, albeit with species-dependent predictability.

An extra cysteine in CDR1 or FR2 is present in ~80% of Bactrian and dromedary camel-derived sdAbs and in ~10% of llama-derived sdAbs, enabling the formation of a non-canonical disulfide bond that is primarily found between CDR1 and CDR3 or FR2 and CDR3. More than 40% of alpaca VHHs contain a non-canonical cysteine within FR2, and the corresponding disulfide bond between FR2 and CDR3 is frequently observed [8,62,63]. In accordance with these previous structural descriptions, in Bactrians and dromedary camels, we observed more frequency of lysines (especially in FR3) and cysteines (CDR1 and CDR3). These noncanonical cysteines in CDR1 and CDR3 can complicate thiol-selective conjugation by reducing the availability of free thiols, increasing product heterogeneity, and risking paratope disruption upon disulfide reduction. Due to the limited number of sdAbs available for this species, these results should be interpreted with caution, as additional data may further refine the observed trends. Although these liabilities can be mitigated (e.g., disulfide rebridging, lysine- or enzyme-mediated labeling, or the introduction of engineered conjugation sites) [64,65], they add substantial complexity. Consequently, for robust, reproducible bioconjugation, alpaca and llama-derived sdAbs (whose cysteine patterns are typically less paratopic or more predictable) remain the most straightforward choice. However, dromedary-derived sdAbs, despite their higher cysteine content, offer valuable structural diversity and functional robustness, making them highly attractive candidates when enhanced stability or unique binding profiles are desired.

In this context, the selection of the source species can be guided by the intended application. Llama and alpaca sdAbs, which display consistent framework composition and predictable cysteine patterns, may be preferred for conjugation and applications requiring high solubility and homogeneity. Conversely, dromedary- and Bactrian-derived sdAbs, characterized by additional cysteines and reinforced disulfide bridges, could be advantageous in contexts demanding enhanced thermostability or structural rigidity.

These results also could provide a valuable source of sdAbs with known targets for AI-assisted sdAb discovery and design. The systematic characterization of framework conservation, loop variability, and residue distribution across species provides an annotated dataset that can be leveraged for data-driven modeling. This information could help extract sequence–structure–function relationships that could serve to train predictive and generative algorithms. By integrating these experimentally validated sdAbs into AI-guided pipelines, it becomes possible to predict key biophysical traits including stability, solubility, or conjugation efficiency, and to design synthetic sdAbs optimized for specific targets or functional constraints.

## 5. Conclusions

We have compiled MO-IISA, a curated collection of camelid sdAbs that enhances existing resources and enables structure-informed comparative analyses. In alignment with the literature, framework regions, particularly FR4, are highly conserved across species. Minor but consistent variations in CDR length were observed between species, with llamas CDR1/2 being the largest. Furthermore, dromedaries showed greater homogeneity in CDR1/2 sizes than llamas and alpacas. Predictable chemical anchor points for bioconjugation are provided by the enrichment of lysines in FR3 (and FR2) and cysteines in FR1 and FR3, with special attention to non-canonical cysteines in dromedary and Bactrian camel-derived CDR1 andCDR3 regions. Collectively, these findings confirm and refine the current understanding of interspecies sdAb diversity and provide practical guidance for labeling strategies, construct design, and species selection in sdAb development.

## 6. Limitations of the Study

*Species coverage and bias*. The number of sdAbs per species is not balanced. There is substantially more llama- and alpaca-derived sdAbs, while data for Bactrian camels are scarce (n = 35) and were excluded from statistical analyses, limiting inferences for that species. Additionally, Bactrian-derived sdAbs originate from a single source database, and it cannot be ruled out that they may derive from the same individual, which limits their representativeness of the species.*Selection by reported targets.* By focusing on sdAbs with known antigen targets or validation, variants present in massive libraries without characterization may be underrepresented, introducing a bias towards more studied antigens and formats. Some antigens in the database are nonspecific (e.g., “human” or referenced in the included DOI) and will require further study. No specific reporting of paratope sequence was provided.*Absence of large-scale structural data.* The study focuses on sequence and derived metrics. It does not systematically integrate resolved 3D structures, quantitative affinities, or stability data for all sdAbs, which limits the ability to establish direct correlations between sequence traits and functional properties.*Uncertain CDR delineation of sdAbs*. A major limitation of our analysis is that a substantial subset of sdAbs was sourced from external databases where the method used to delineate CDRs is unknown. This lack of traceability can introduce numbering heterogeneity (e.g., IMGT, Kabat, Chothia, AHo), errors at FR/CDR boundaries, and consequent biases in conservation metrics, length estimates, and variability maps.*Generalization of data.* Implications for bioconjugation (e.g., Lys/Cys accessibility) are inferred from average distributions; individual cases, particularly those containing non-canonical cysteines in CDRs, require specific experimental validation to confirm their specific reactivity and impact.

## Figures and Tables

**Figure 1 antibodies-14-00100-f001:**
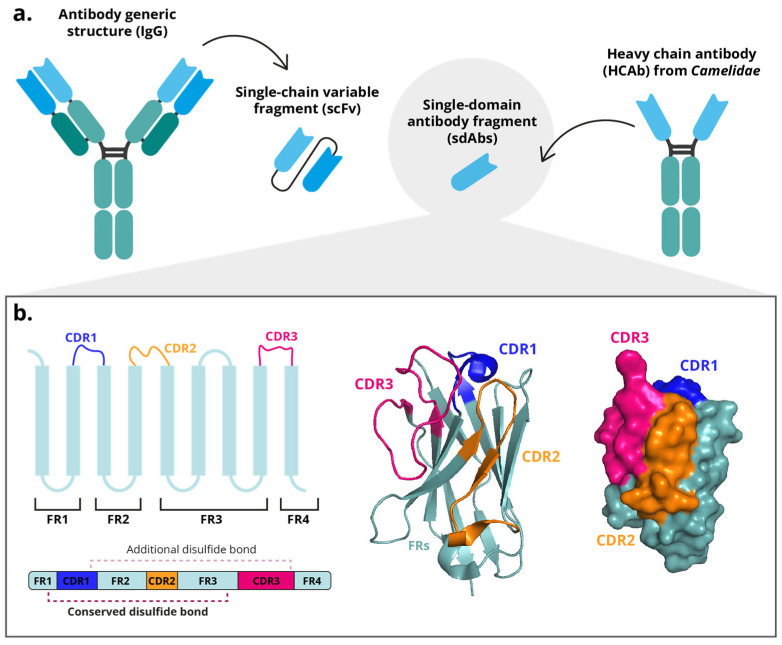
Structure of a single-domain antibody fragment (sdAb). (**a**) Diagram of a conventional antibody (IgG) with a single-chain variable fragment (scFv) and a heavy chain antibody (HCAb) with a sdAb. (**b**) Schematic representation of the VHH B-strands and their 3D representation from the PDB structure 5u65.

**Figure 2 antibodies-14-00100-f002:**
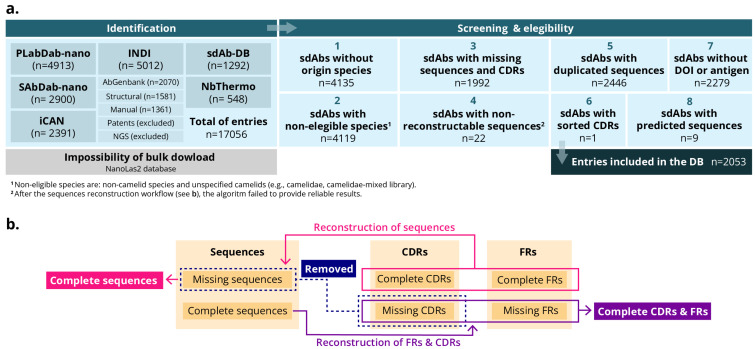
Schemes of the workflow for the MO-IISA database design. (**a**) Identification, screening, eligibility criteria and inclusion of the entries for the database. (**b**) Sequence workflow.

**Figure 3 antibodies-14-00100-f003:**
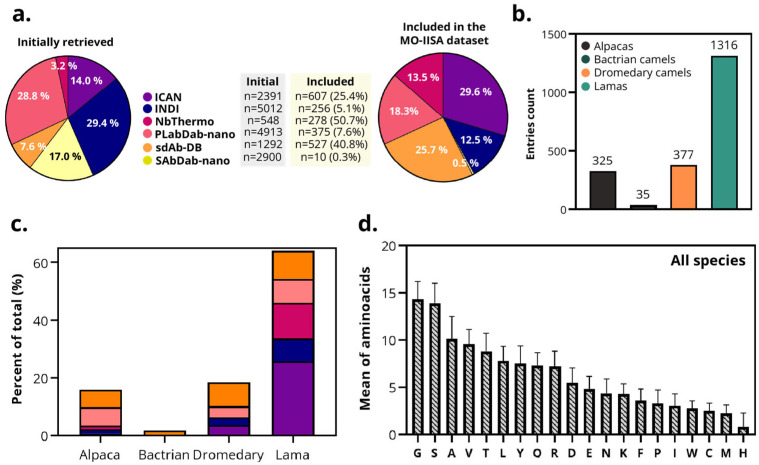
Overview of the MO-IISA database. (**a**) Initial sdAbs entries collected from each database (iCAN, INDI, SAbDab-nano, sdAb-DB, PLabDab-nano, and NbThermo) before and after screening and application of eligibility criteria. Color legend of the different databases is defined here for the rest of the panels. (**b**) Number of sdAbs retrieved from each dataset by species of origin (alpacas, Bactrians, dromedaries, and llamas). (**c**) Proportion of sdAbs retrieved from each database by species, normalized for the total sdAbs of each species. (**d**) Average frequency distribution of the 20 amino acids across all sdAbs sequences included in the MO-IISA database.

**Figure 4 antibodies-14-00100-f004:**
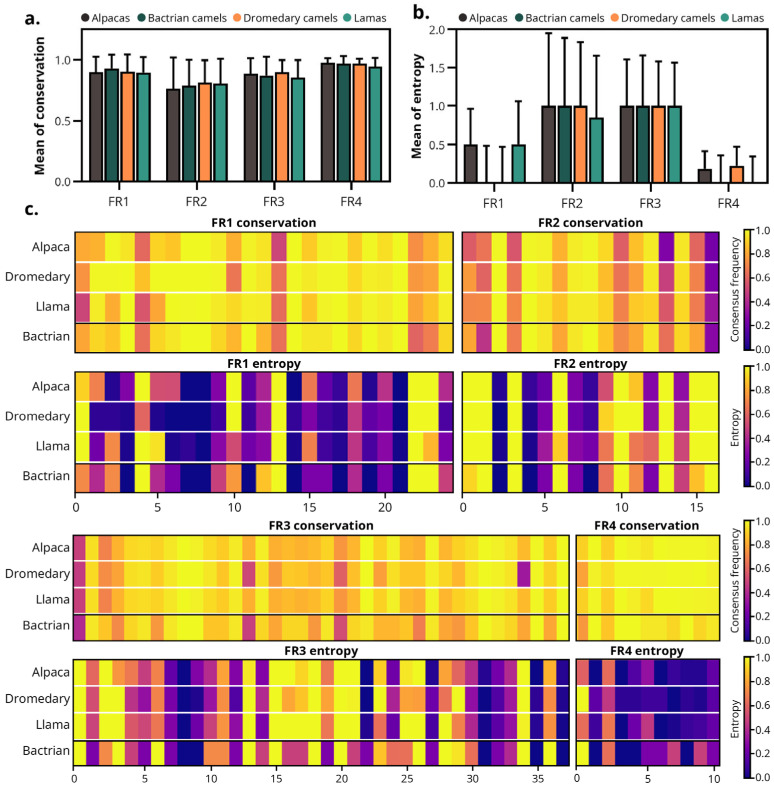
Frameworks conservation and entropy of the camelid-derived sdAbs. (**a**) Mean conservation and (**b**) entropy of the FRs (FR1–FR4) of sdAbs from all species included in the study. (**c**) Heatmaps showing the conservation and the entropy of each amino acid position within the FRs. Color scale: purple to yellow, indicates low to high values—conservation panels show consensus frequency (low = poorly conserved, high = highly conserved), while entropy panels represent Shannon entropy (low = low diversity, high = high diversity). Note: Due to the small sample size of Bactrians (n = 35), data from this species were excluded from statistical studies and are shown depicted separated by a line.

**Figure 5 antibodies-14-00100-f005:**
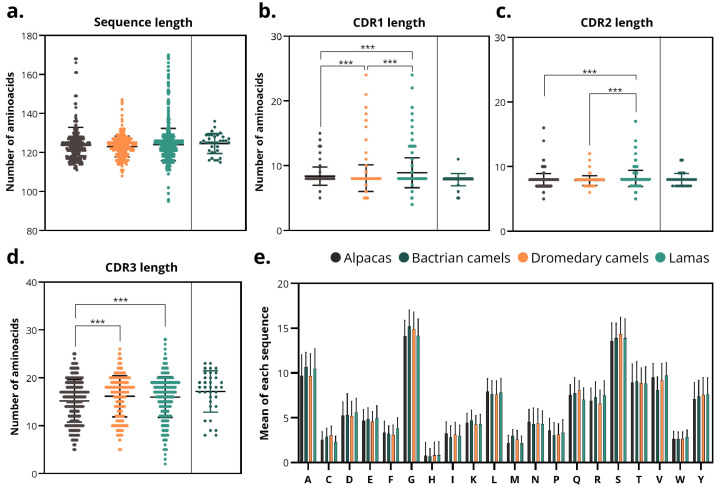
Sequence and CDR length analysis. (**a**) Total sdAbs sequence length across species of origin. (**b**–**d**) Lengths of (**b**) CDR1, (**c**) CDR2 and (**d**) CDR3 regions in sdAbs from different species. Asterisks denote significance thresholds for the Kruskal–Wallis and pairwise Mann–Whitney tests (Holm-adjusted *p*): *p* < 0.001 (***). (**e**) Average individual amino acid composition of sdAbs sequences by species. Note: Due to the small sample size of Bactrian camel sdAbs (n = 35), data from this species, despite being depicted in the figure, were excluded from statistical studies.

**Figure 6 antibodies-14-00100-f006:**
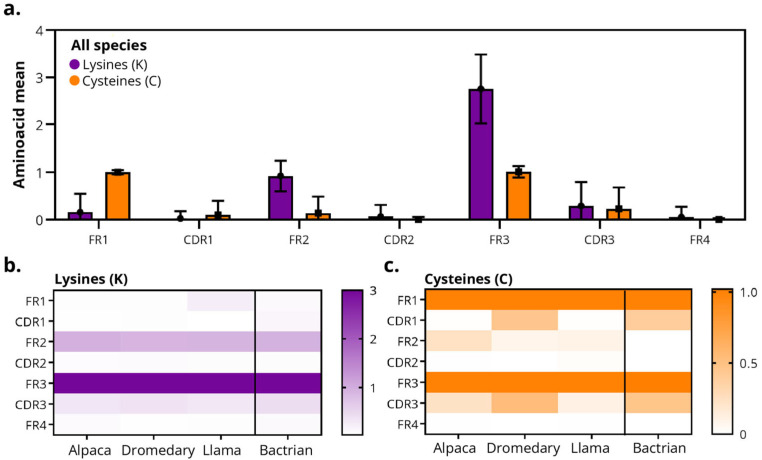
Distribution of lysines (K) and cysteines (C) residues in camelid sdAbs. (**a**) Average frequency of lysines (K, purple) and cysteine (C, orange) per region of the sdAb sequence (FR1–FR4 and CDR1–CDR3) across all species. (**b**,**c**) Heatmaps showing (**b**) lysine and (**c**) cysteine distribution by region and species. Color scales indicate the fractional frequency (0–1) of residue presence per region. In all species, the canonical cysteines of the immunoglobulin fold are conserved in FR1 and FR3, with low incidence in CDRs.

## Data Availability

The original data presented in the study are openly available in GitHub at https://github.com/Javiersp4/MO-IISA (accessed on 6 October 2025).

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
