# Peer review of "Species-Dependent Structural Variations in Single-Domain Antibodies"

_2073-4468, 2025, doi:10.3390/antib14040100_

Round 1

Reviewer 1 Report

Comments and Suggestions for Authors

This is a thorough and carefully executed study that will be very interesting to anyone working with these VHH antibodies

Reviewer 2 Report

Comments and Suggestions for Authors

General comments: The authors state that (lines 120-121) features such as stability, solubility and potential biomedical application of sdAbs are influenced by species-specific sequences. While these could be quite novel findings in a large-scale sequence analysis, a conclusion related to these criteria and the sequences analyzed is not spelled out clearly in the results and discussion. Although there have been some discussions in regard to using Lysin residues in the FR3 for conjugation or the use of non-cannonical cyctein for potential conjugation, however, readers expect a clear conclusion in choosing either species for isolating sdAbs for certain applications. Under the current format, the discussion around this topic is weak and needs to be strengthened.

Discussion around the CDR1/H1 is a bit controversial; on one hand, authors talk about the limitation and restriction of diversity and length, and on the other hand, it is argued that extended length and variability of the H1 is a way of compensation for the lack of the light chain. Obviously, the latter argument has already been demonstrated before throughout the literature

I also suggest that authors review the manuscript text and remove a relatively good number of sentences which are unclear or repeated. Additionally, special attention is needed for the reference cited in the text, as a few seem to be irrelevant to the topic discussed.

Specific comments:

  1. Line 10: suggest adding “camelid” in the sentence “from camelid heavy-chain…”
  2. Line 55: suggest rewriting “sdAb can be synthetically….in microorganism.”
  3. Line 57: “after immunization….”. Based on the literature, the total lymphocytes, including the ones producing conventional camelid antibodies, are isolated and not “HCAb producing”.
  4. Line 67: The CDRs or of a VHH….”
  5. Line 68: suggest rewriting: “....are spatially grouped with the N-terminus”.
  6. Line 70: The concept of H loops and CDRs needed to be defined as these two were used interchangeably, while the exact concept is not the same in terms of amino acid numbering. Then, in line 70,  they talk about longer H1 and H3, and in lines 72-73, they talk about H1 and H2 length to be restrictive and conserved. These seem to contradict each other.
  7. Line 75-76: the percentage of homology mentioned here was not found in the references 24 and 25, and these two references seem to be irrelevant in this context. Most references are talking about similarities of 80% or higher with the human VH family, and if the authors found otherwise, please mention it with the right reference.
  8. Lines 81-83: consider adding the IMGT number for the position of amino acids, as in the current format, it is not clear which number system has been applied.
  9. Line 121: I suggest that authors relate their results to the specific aim of the studies, namely, how the results clearly define each species’ VHH has an identifier sequence pattern that influences the stability, solubility, and potential for biomedical application.
  10. Lines 238-239: The two sentences seem to be repeated when talking about the role of cysteine and lysine residues.
  11. Lines 323- 327: the first sentence (lines:)323-324) and last sentence (lines: 325-327) seem to be repeated.
  12. Lines 375: The percentage mentioned here was not found in the cited references. The references seem to be irrelevant here.
  13. Lines 389: Please cite the related reference after “ consistent with the literature (Ref?)….”
  14. Lines 421-422: repetition of “high” seems irrelevant and needs rewriting.
  15. Lines 430-431: The sentence seems to be unclear and needs rewriting.
  16. Lines 468-469: The sentence does not read well, and something is missing here.
Comments on the Quality of English Language

I think the quality of English needs to be improved.

Reviewer 3 Report

Comments and Suggestions for Authors

The authors assembled a dataset MO-IISA comprising over 2000 sdAbs with known targets from various species and provided comparative analysis. Overall, the description of the freely available dataset and discussion are thorough. Here are some suggestions.

Figure 3A: Please specify what does each pie chart refer to.

Line 303: The expression of "slightly but significantly" can be ambiguous. Please use "significantly" if you would like to refer to statistic differences.

Line 304: As shown in Figure 4B, the CDR1 length (please correct "lenght" in Figure 4B, 4C and 4D) showed significant difference among some species. Please provide the statistic rationale for the statement that CDR1s were more homogeneous in size.

Please check the Figure 4 legend and align the description with different panels.

In some places in the text, “lysines” were written as “lysins”. E.g., line 354.

Conclusion, lines 494 - 495: According to results in section 3.2, the length of llama CDRs were not the shortest.

Comments on the Quality of English Language

There are some minor issues with grammar and expression that can be improved.

Round 2

Reviewer 3 Report

Comments and Suggestions for Authors

The comments and suggestions have been well addressed.